# Knee joint distraction in regular care for treatment of knee osteoarthritis: A comparison with clinical trial data

Mylène P. Jansen[1]*, Simon C. Mastbergen[1], Ronald J. van Heerwaarden[2], Sander Spruijt[3], Michelle D. van Empelen[4], Esmee C. Kester[4], Floris P. J. G. Lafeber[1]⊙, Roel J. H. Custers[4]⊙

**1** Department of Rheumatology & Clinical Immunology, University Medical Center Utrecht, Utrecht, The Netherlands, **2** Centre for Deformity Correction and Joint Preserving Surgery, Kliniek ViaSana, Mill, The Netherlands, **3** Department of Orthopedics, HagaZiekenhuis, Den Haag, The Netherlands, **4** Department of Orthopedic Surgery, University Medical Center Utrecht, Utrecht, The Netherlands

⊙ These authors contributed equally to this work.

* m.p.jansen-36@umcutrecht.nl

**Data Availability Statement:** Data related to this manuscript cannot be shared publicly because of ethical restrictions related to participant consent.

## Abstract

### Objectives

Knee joint distraction (KJD) has been evaluated as a joint-preserving treatment to postpone total knee arthroplasty in knee osteoarthritis patients in three clinical trials. Since 2014 the treatment is used in regular care in some hospitals, which might lead to a deviation from the original indication and decreased treatment outcome. In this study, baseline characteristics, complications and clinical benefit are compared between patients treated in regular care and in clinical trials.

### Methods

In our hospital, 84 patients were treated in regular care for 6 weeks with KJD. Surgical details, complications, and range of motion were assessed from patient hospital charts. Patient-reported outcome measures were evaluated in regular care before and one year after treatment. Trial patients (n = 62) were treated and followed as described in literature.

### Results

Patient characteristics were not significantly different between groups, except for distraction duration (regular care 45.3±4.3; clinical trials 48.1±8.1 days; p = 0.019). Pin tract infections were the most occurring complication (70% regular care; 66% clinical trials), but there was no significant difference in treatment complications between groups (p>0.1). The range of motion was recovered within a year after treatment for both groups. WOMAC questionnaires showed statistically and clinically significant improvement for both groups (both p<0.001 and >15 points in all subscales) and no significant differences between groups (all differences p>0.05). After one year, 70% of patients were responders (regular care 61%, trial 75%; p = 0.120). Neither regular care compared to clinical trial, nor any other characteristic could predict clinical response.

These restrictions are imposed by the institutional review board of the University Medical Center Utrecht, Utrecht, The Netherlands. All relevant data are available upon request by sending an email to the Rheumatology department of the UMC Utrecht (urrci@umcutrecht.nl).

**Funding:** The institution of MP Jansen, SC Mastbergen and FPJG Lafeber has, during the study period, received funding from: - Dutch Arthris Society (ReumaNederland), project number LLP-9, https://reumanederland.nl/. - ZonMW (The Netherlands Organization for Health Research and Development), project number 95110008, https://www.zonmw.nl/. The funders had no role in study design, data collection and analysis, decision to publish, or preparation of the manuscript.

**Competing interests:** FPJG Lafeber is co-founder, co-director, and shareholder of ArthroSave BV, a medical device company involved in marketing a user-friendly knee joint distraction device. This does not alter our adherence to PLOS ONE policies on sharing data and materials.

## Conclusions

KJD as joint-preserving treatment in clinical practice, to postpone arthroplasty for end-stage knee osteoarthritis patient below the age of 65, results in an outcome similar to that thus far demonstrated in clinical trials. Longer follow-up in regular care is needed to test whether also long-term results remain beneficial and comparable to trial data.

## Introduction

Knee osteoarthritis (OA) is characterized by articular cartilage degeneration and is an important cause of pain and disability in adults.[1,2] While total knee arthroplasty (TKA) is a widely accepted intervention for end-stage knee OA, it poses a major healthcare burden when placed in younger patients, since they have a higher risk of needing a costly and less effective revision surgery later in life.[3–6]

Knee joint distraction (KJD) is a joint-preserving treatment for knee OA for younger patients, where the knee joint is temporarily fully unloaded by distraction of tibia and femur, using an external fixation frame.[7] In an open prospective study (OPS) between 2006 and 2008, twenty knee OA patients below the age of 60, indicated for TKA were treated for eight weeks with KJD.[8] These patients showed long-term, in the first two years progressive, significant clinical benefit and cartilage tissue regeneration. In over three quarters of the patients, TKA could be postponed for over five years, and half of the patients was still without prosthesis nine years after treatment.[8–11] After this trial the distraction period was shortened to six weeks, as this was considered sufficient.[12] Between 2011 and 2014, the six-week KJD was studied in comparison to TKA or to high tibial osteotomy (HTO) in two separate randomized controlled trials (RCTs). In both trials combined, 41 KJD patients gained significant clinical and structural benefit in the first year, which was shown to be maintained up to at least two years after treatment. Both trials demonstrated that KJD was non-inferior to the alternative treatment.[13–15] Since 2014, KJD is offered as a regular care treatment in a limited number of hospitals for knee OA patients under the age of 65.

Often when a new treatment proceeds from clinical trial to regular care, indications for treatment broaden and treatment outcome weakens. As such, treatment and surgery details, baseline characteristics, complications during treatment, and treatment efficacy of KJD in regular care were compared with clinical trial (OPS/RCT) conditions.

## Methods

### Patients

In regular care, at the department of Orthopedic Surgery in our hospital patients are offered KJD in case they are considered for TKA but still younger than 65. According to local guidelines for treating patients with TKA, patients have had sufficient conservative treatment, but with insufficient success and a Kellgren-Lawrence grade (KLG) of at least 2. Patients with presence or history of inflammatory joint condition, joint prosthesis elsewhere in the body (potential risk of prosthetic joint infection), or physical or social conditions that do not support a six-week distraction period, are ineligible. The standard procedure at the department of orthopedics is that patients are asked for consent to use their anonymized data for future research purposes, which all patients in the present study provided. Official ethical approval was ruled as

not required by the medical ethical review committee of the University Medical Center Utrecht (protocol number 17-005C) and all patients give written informed consent.

In the open prospective study (OPS) and the two randomized controlled trials (RCTs), inclusion criteria were: medial tibio-femoral compartmental OA; intact knee ligaments; normal range-of-motion (min. of 120˚ flexion); normal stability; BMI <35; Visual Analogue Scale of pain ≥60 mm, radiographic signs of joint damage and tibiofemoral OA (radiological joint damage KLG>2 as judged by the orthopedic surgeon). Exclusion criteria were (among others): presence or history of inflammatory or septic arthritis; severe knee malalignment (>10˚) requiring surgical correction; psychological inabilities or difficult to instruct; joint prosthesis elsewhere in the body; not able to undergo MRI examination; post traumatic fibrosis due to fracture of the tibial plateau; surgical treatment of the involved knee <6 months ago; contra-lateral knee OA that needs treatment; primary patello-femoral OA. For the OPS the age was <60 years, for the RCTs <65 years. For the OPS and RCT *versus* TKA, all patients had to be considered for TKA. For the RCT *versus* HTO, all patients had to be considered for HTO, with medial compartmental knee OA with a varus deviation of <10˚. All inclusion aspects have been described in detail for all three studies, previously.[8,13,14,16] All trials were granted ethical approval by the medical ethical review committee of the University Medical Center Utrecht (protocol numbers 04/086, 10/359/E, and 11/072) and registered in the Netherlands Trial Register (trial numbers NL419, NL2761 and NL2680). All patients gave written informed consent.

## Knee joint distraction treatment

KJD was performed by fixating an external distraction device to the femur and tibia using eight half pins according to a standardized surgical procedure. In all patients, a device was used consisting of two distraction tubes with internal springs, one placed medially and one laterally of the knee joint (Fig 1). The half pins (self-drilling, 5 mm diameter) used to fixate the distraction tubes were placed in pairs at four different locations (tibia/femur and medial/lateral), all placed outside the knee joint area to prevent complications during a potential future prosthesis surgery. [17] The medial femoral pins were positioned parallel to the knee joint line in an approximately 10˚ dorsomedial—ventrolateral direction (10˚ angulation to the frontal plane) to minimize interference of the half pins with the quadriceps muscles. The lateral femoral pins were placed parallel to the knee joint line, perpendicular to the tibial bone axis, and approximately in the frontal plane. The medial tibial half pins were positioned parallel to the knee joint space, and if possible perpendicular to tibial bone axis and the anteromedial tibial face, approximately at 35˚ to the frontal plane. The lateral tibial half pins used the same slope of approximately 35˚ to the frontal plane. Proper positioning and depth, with slight protrusion of the half pin (of the pointed tip only) through the second cortex, was checked using fluoroscopy (C-arm). After positioning the half pins and distraction tubes, according to standardized surgical procedures, a distraction distance of 2 mm was provided intra-operatively. All this was performed under general or spinal anesthesia, depending on the surgeon's and patient's preference.

## In regular care

In regular care, the average intervention time (the time between the first incision and the surgeon being finished) was 53 (range 31–79) minutes. Blood loss during surgery was in all cases negligible. After surgery, patients generally stayed in the hospital for another two to three days, during which the tubes were gradually distracted until 5 mm distraction was reached. At completion, the distraction distance was checked on weight-bearing radiographs and adapted if needed. During the distraction period weight-bearing, supported with crutches if needed, was allowed and encouraged. This provides intra-articular fluid pressure changes, considered

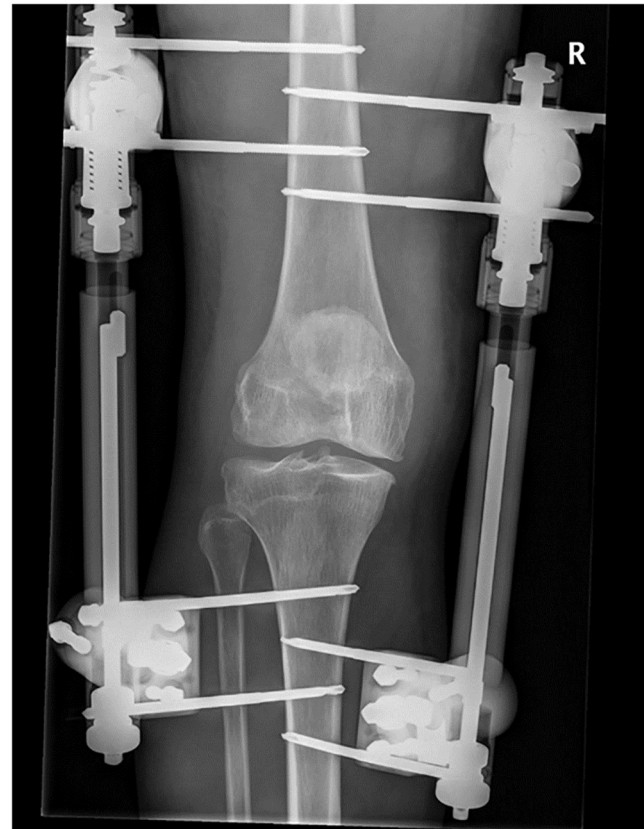

**Fig 1. Representative radiograph of the external distraction frame in use.**

relevant for nutrition of the cartilage, because of 3 mm axial displacement under 80 kg of weight-bearing of the internal springs. [18,19] Patients received low molecular weight heparin for six weeks and a standard prescription for seven days of oral antibiotics (flucloxacillin). If patients suspected a pin tract infection, based on consulting their physician, a course of flucloxacillin was started. During the distraction period, patients visited the outpatient clinic once for a general evaluation. After six weeks, the distraction frame was removed and knee manipulation (flexion-extension) was performed under general or spinal anesthesia at day-treatment. The total frame removal time in regular care was 16 minutes (range 7–36) and patients were discharged the same day.

## Under trial conditions

The above described treatment was used for all patients included in the RCTs as well. However, the patients treated in the OPS received eight instead of six weeks of distraction and returned to the hospital every two weeks, where the tubes were temporarily removed and the knee was flexed and extended by use of continuous passive motion device for three to four hours. Pain at the pin sites determined the maximum degree of flexion (average 25˚; range 15˚-80˚).[8]

## Follow-up

In regular care, weight-bearing PA radiographs were taken and the range of motion (ROM) was measured pre-surgery and at four and twelve months after frame removal in the

outpatient clinic. A standard registry for all orthopedic patients provided data on patient-reported outcome measures (PROMs). Patients were requested to fill out several PROMs by questionnaires, before surgery and three, six, and twelve months after surgery, and every year thereafter. This is done automatically by e-mail, without reminder, causing relatively high numbers of missing data.

Trial patients were seen at comparable time points (six and twelve months after frame placement) where the ROM was measured and questionnaires were filled out on paper, causing limited missing data. One-year follow-up results have been published previously for each trial separately.[8,13,14]

No standardized radiographs were made in regular care and for that reason in clinical practice, the in previous trials reported cartilaginous tissue repair could only be confirmed quantitatively. Since this outcome is a major benefit of the distraction treatment, two representative sets of pre- and one year post-treatment radiographs of a regular care patient and clinical trial patient have been provided.

## Data collection

All regular care KJD patients treated in our hospital before 2018 were included and thus provided one-year follow-up. Electronic charts of these patients were evaluated to check essential baseline characteristics. The ROM, measured by the orthopedic surgeon, and complications as a result of treatment had been registered for these patients, data which was also available from the OPS and RCT patients.

Only data collected for both regular care and clinical trial patients were compared. The Western Ontario and McMaster Universities Osteoarthritis Index (WOMAC, version 3.1) questionnaire was used for evaluation of clinical efficacy, as this questionnaire was available for all patient groups. Since regular care patients filled out their questionnaires online, a relatively large amount of missing data is expected. To limit bias, only patients who filled out the questionnaires both before and one year after treatment were included in the analysis of clinical efficacy, and characteristics of these patients were compared to the entire group of regular care patients.

## Statistical analysis

Characteristics were compared between regular care and clinical trial patients using independent t-tests or, in case of categorical variables, chi-square tests. WOMAC data before and one year after treatment was compared for both groups separately, using paired samples t-tests. The one-year WOMAC values were compared and tested between groups for clinical significance, defined as a difference of more than 15 WOMAC points [20], and for statistical significance using linear regression, corrected for baseline values and possible significantly different baseline or treatment characteristics. The influence of different baseline characteristics on the one-year change in total WOMAC score, corrected for baseline WOMAC, was identified using linear regression. Being a responder to KJD treatment was analyzed according to the Outcome Measures in Rheumatology-Osteoarthritis Research Society International (OMERACT-OARSI) responder criteria, defined as an increase of $\geq 50\%$ and $\geq 20$ points in WOMAC pain or function scales, or a $\geq 20\%$ and $\geq 10$-point improvement in both scales, and potential predictors identified.[21]

For all values, mean and standard deviations (SD) are given, and for all changes over time the mean change and 95% confidence interval (95%CI) are shown. P-values <0.05 were considered statistically significant. IBM SPSS Statistics version 25 (IBM Corp; Armonk, NY) was used for all statistical analyses.

**Table 1. Baseline characteristics of patients treated with knee joint distraction in regular care and in clinical trials.**

| Baseline characteristics, mean ± SD or n (%) | Regular care (n = 84) | Clinical trial (n = 62) | p-value |
|---|---|---|---|
| Age (years) | 53.1 ± 6.9 | 51.5 ± 6.9 | 0.173 |
| Male gender | 52 (62) | 36 (58) | 0.639 |
| BMI (kg/m$^2$) | 27.9 ± 3.7 | 28.2 ± 3.7 | 0.639 |
| Left index knee* | 43 (51) | 26 (42) | 0.268 |
| Range of motion (degrees) | 124.2 ± 17.8 | 122.7 ± 14.7 | 0.602 |
| Leg axis (degrees) | 4.3 ± 5.1 | 4.9 ± 4.4 | 0.556 |
| Varus/valgus* | 57 (68) / 16 (19) | 28 (45) / 3 (5) | 0.140 |
| Kellgren-Lawrence grade* | | | 0.401 |
| - Grade 0 | 0 (0) | 0 (0) | |
| - Grade 1 or 2 | 19 (23) | 18 (29) | |
| - Grade 3 or 4 | 64 (76) | 44 (71) | |
| Distraction duration (days) | 45.3 ± 4.3 | 48.1 ± 8.1 | **0.019** |

P-values of continuous variables are calculated with independent t-tests and for categorical variables with chi-square tests (indicated with *). Bold p-values indicate statistical significance.

## Results

### Baseline characteristics

Before 2018, 84 patients were treated with KJD in regular care in our hospital and all accepted to participate in the orthopedic standard registry. Between 2006 and 2014, 62 patients were treated in the three trials combined. The baseline characteristics of both groups are shown in Table 1, showing a different distraction duration between both groups, which was longer for clinical trial patients (48.1±8.1 days; regular care 45.3±4.3; p = 0.019), but shorter when excluding the OPS patients who received distraction for eight instead of six weeks (RCT 42.8±2.3; regular care 45.3±4.3; p<0.001).

In one patient in the regular care group compartment syndrome occurred and the distraction frame was removed after two days. This patient was excluded from the distraction duration in Table 1, since no full treatment was applied.

### Cartilaginous tissue repair

Radiographs of a representative regular care patient and a trial patient pre-treatment and one year post-treatment are shown in Fig 2. In both cases, despite the absence of quantification of the joint space widening in clinical practice, a clear increase in joint space width is demonstrated, in previous studies clearly related to cartilage thickening using MRI and biochemical markers.[9,10,15]

### Complications

All treatment-related complications that occurred are summarized in Table 2. Pin tract infections occurred most often and in 86% of cases were successfully treated with oral antibiotics. A combination of intravenous and oral antibiotics was necessary in 14% of pin tract infections. OPS patients had significantly more pin tract infections than RCT patients (OPS 85%; RCT 57%; p = 0.030). There was no significant difference in pin tract infections between regular care patients and any of the trial patient groups (OPS/RCT, OPS or RCT; all p>0.1). Patients experiencing osteomyelitis (six patients) were treated with additional surgical cleaning of pin tract wounds and a combination of intravenous (2 weeks) and oral (4 weeks) antibiotics

according to a local standardized treatment protocol for osteomyelitis. Pin loosening (three patients) or breaking (one patient, reason unknown) was treated by tightening or refixation of the pins at either the emergency room or the outpatient clinic, while the one patient experiencing pin tract bleeding received a pressure bandage at the emergency room. Both deep venous thrombosis (two patients) and pulmonary embolisms (three patients) were treated with extra anticoagulation, which in case of a pulmonary embolism included hospitalization. For the patient experiencing a suspected compartment syndrome, the frame was immediately removed and a fasciotomy was performed, while the one patient who had pneumonia received intravenous antibiotics.

Of patients with complications, fifteen experienced them after frame removal. Ten were post-distraction infections, treated with oral antibiotics (three patients) or a combination with intravenous antibiotics (seven patients), and one was a post-operative foot drop, successfully treated with an ankle-foot orthosis. The cause has been discussed previously.[14] Flexion

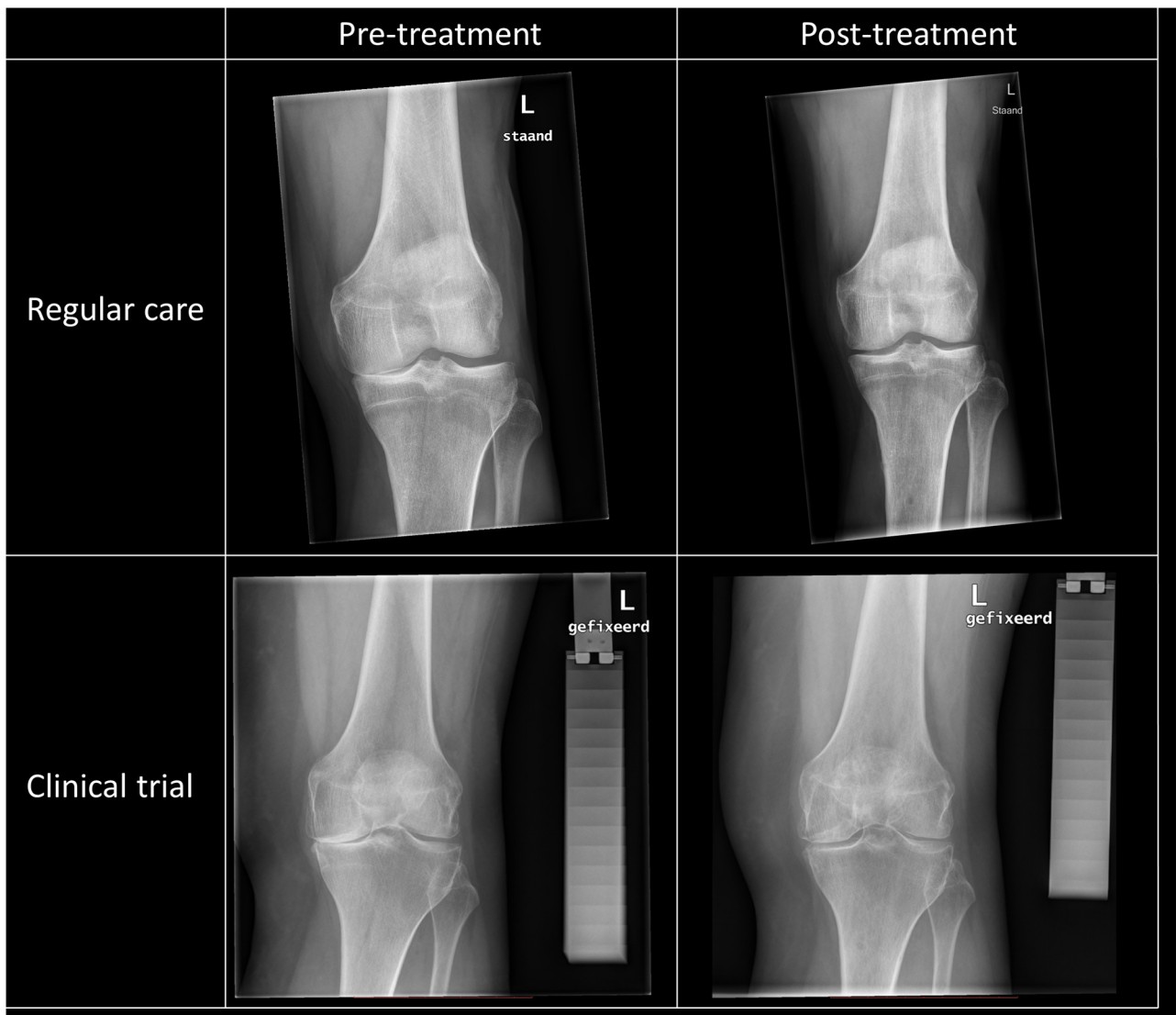

**Fig 2. Representative radiographs pre-treatment and one year post-treatment for regular care and clinical trial patient.** Note the aluminum step wedge needed for joint space width quantification as used in clinical trials.

**Table 2. Complications during and after treatment with knee joint distraction in regular care and in clinical trials.**

| Complications, n (%) | Regular care (n = 84) | Clinical trial (n = 62) |
|---|---|---|
| Pin tract skin infection | 59 (70) | 41 (66) |
| - Oral antibiotics | 51 (61) | 35 (56) |
| - Hospital admission + intravenous antibiotics | 8 (10) | 6 (10) |
| Osteomyelitis | 5 (6) | 1 (2) |
| - Confirmed osteomyelitis | 2 (2) | 1 (2) |
| - Infection treated as osteomyelitis | 3 (4) | 0 (0) |
| Pin loosening | 4 (5) | |
| Flexion limitation | 2 (2) | 1 (2) |
| Deep venous thrombosis | 2 (2) | |
| Pulmonary embolism | 1 (1) | 2 (3) |
| Pin tract bleeding | 1 (1) | |
| Compartment syndrome | 1 (1) | |
| Pneumonia | 1 (1) | |
| Corpus liberum | 1 (1) | |
| Post-operative foot drop | | 1 (2) |
| Breaking of bone pin | | 1 (2) |

limitation (three patients) was treated with manipulation under anesthesia and in one case arthroscopic arthrolysis, while the corpus liberum (a loose piece of cartilage/bone) present in one patient after treatment was arthroscopically removed.

The decrease in ROM shortly after distraction as observed in regular care (-26.5˚; 95%CI -32.0 –-21.0; p<0.001) and the clinical trials (-20.1˚; 95%CI -26.6 –-13.6; p<0.001) was largely regained within four months. Compared to baseline ROM, the regular care patients showed a statistically significant decrease at four months (-5.8˚; 95%CI -10.2 –-1.4; p = 0.011), but not at twelve months (-2.3˚; -6.3–1.8; p = 0.263), as shown in Fig 3. Clinical trial patients showed no statistically significant difference at four months (-3.5˚; -7.4–0.5; p = 0.085) and twelve months (+2.7˚; -0.6–6.0; p = 0.112). When correcting for baseline ROM and distraction duration, there was a statistically significant difference between regular care and clinical trial patients for the twelve-month change (p = 0.013), but not the four-month change (p = 0.232).

## Clinical benefit

In total 41 regular care patients and 61 clinical trial patients completed both baseline and one-year follow-up WOMAC questionnaires, 43 regular care patients were missing because they did not respond to the electronic requests to fill out the questionnaires by E-mail. One RCT patient was missing at one-year follow-up after undergoing additional treatment. The baseline characteristics of the patients who completed both WOMAC questionnaires are shown in Table 3, showing a significant difference only in distraction duration, which again was longer for clinical trial patients (48.2±8.2 days; regular care 45.5±4.2; p = 0.032), but shorter when excluding the OPS patients (RCT 42.8±2.3; regular care 45.5±4.2; p = 0.001).

No statistical significant differences between the 43 regular care patients without and 41 patients with 1 year follow-up data were observed.

As shown in Table 4 and Fig 4, the total WOMAC (Fig 4A) and pain (Fig 4B), stiffness (Fig 4C), and function (Fig 4D) subscales increased statistically and clinically significantly for the 41 regular care patients and 61 clinical trial patients that completed the questionnaires (all p<0.001). Although there was a tendency towards better results for the clinical trial patients,

## Range of Motion

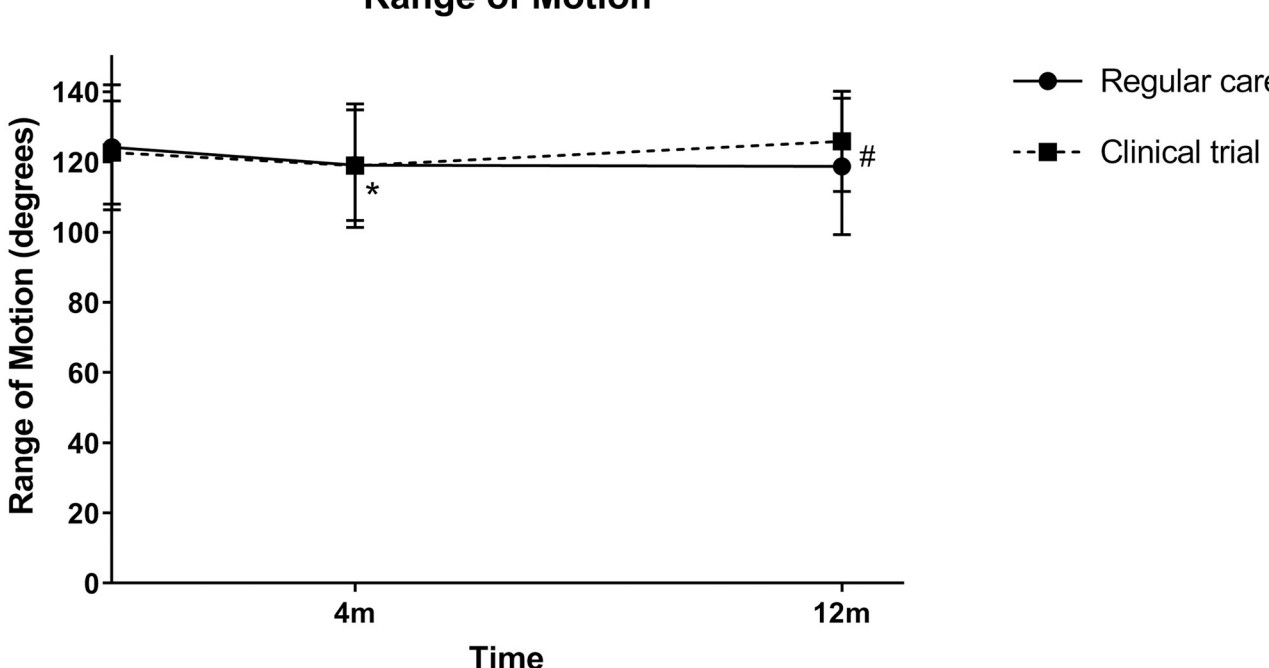

**Fig 3. Range of motion before and after treatment with knee joint distraction.** Statistically significant differences compared to baseline are indicated with * for regular care patients (non-existent for clinical trial patients); statistically significant differences between regular care and clinical trial patients are indicated with #.

**Table 3. Baseline characteristics of patients treated with knee joint distraction in regular care and in clinical trials, who completed both WOMAC baseline and 12-month follow-up questionnaires.**

| Baseline characteristics, mean ± SD or n (%) | Regular care (n = 41) | Clinical trial (n = 61) | p-value |
|---|---|---|---|
| Age (years) | 54.0 ± 6.9 | 51.7 ± 6.8 | 0.102 |
| Male gender* | 23 (56) | 35 (57) | 0.898 |
| BMI (kg/m²) | 27.5 ± 3.9 | 28.1 ± 3.7 | 0.508 |
| Left index knee* | 19 (46) | 26 (43) | 0.711 |
| Range of motion (degrees) | 125.4 ± 14.1 | 122.7 ± 14.9 | 0.362 |
| Leg axis (degrees) | 4.6 ± 4.7 | 4.8 ± 4.4 | 0.879 |
| Varus/valgus* | 33 (80) / 6 (15) | 27 (44) / 3 (5) | 0.510 |
| Kellgren-Lawrence grade* | | | 0.152 |
| - Grade 0 | 0 (0) | 0 (0) | |
| - Grade 1 or 2 | 7 (17) | 18 (30) | |
| - Grade 3 or 4 | 34 (83) | 43 (70) | |
| Distraction duration (days) | 45.5 ± 4.2 | 48.2 ± 8.2 | **0.032** |
| WOMAC Total | 47.5 ± 14.9 | 49.8 ± 15.7 | 0.464 |
| WOMAC Pain | 46.3 ± 16.9 | 49.8 ± 15.7 | 0.293 |
| WOMAC Stiffness | 39.3 ± 23.1 | 45.4 ± 18.3 | 0.141 |
| WOMAC Function | 48.9 ± 15.2 | 51.0 ± 16.2 | 0.498 |

P-values of continuous variables are calculated with independent t-tests and for categorical variables with chi-square tests (indicated with *). Bold p-values indicate statistical significance. WOMAC = Western Ontario and McMaster Universities Osteoarthritis Index.

**Table 4. Clinical outcome for patients treated with knee joint distraction in regular care and in clinical trials.**

| ΔWOMAC, mean (95%CI) | Regular care (n = 41) | Clinical trial (n = 61) | P-value |
|---|---|---|---|
| Total | 22.2 (15.1–29.3)* | 28.3 (23.5–33.1)* | p = 0.080 |
| Pain | 24.0 (16.2–31.9)* | 29.5 (24.2–34.7)* | p = 0.104 |
| Stiffness | 20.4 (11.2–29.7)* | 19.5 (12.9–26.1)* | p = 0.463 |
| Function | 21.9 (14.8–29.0)* | 28.6 (23.7–33.6)* | p = 0.069 |

WOMAC = Western Ontario and McMaster Universities Osteoarthritis Index. Significant one-year changes are indicated with * while the p-values indicate differences in one-year changes between regular care and clinical trial patients, calculated with linear regression, corrected for baseline WOMAC and distraction duration. 95%CI = 95% confidence interval.

no clinically or statistically significant differences in one-year changes between regular care and trial patients were observed (all p>0.068). Similar data were found for OPS and RCT patients separately, although for OPS patients slightly, but not statistically significantly, better results were obtained.

After one year, 70% of patients were OMERACT-OARSI responders (regular care 61%, clinical trial 75%; p = 0.120).

Neither regular care versus trial treatment nor any of the other baseline characteristics had a significant influence on the one-year change in total WOMAC score, neither in univariable nor multivariable models, or on being a responder. Experiencing pin tract infections or complications in general did not have a significant influence on one-year WOMAC change or being a responder (all p>0.2).

## Discussion

Knee joint distraction is a relatively new, joint-preserving treatment for knee OA that after several clinical trials is now used in clinical practice to postpone a first TKA. This enabled evaluation if patients treated in regular care still have a similar indication profile, viz. similar characteristics as those treated in clinical trials and if KJD is still as clinically effective in regular care as it was shown to be in the trials.

Despite the fact that regular care usually does not use selection criteria as strictly as clinical trials do, this study showed that the 84 patients treated with KJD in regular care between 2014 and 2018 had in general the same characteristics as the patients included in clinical trials the years before. Only the distraction duration was shorter in the regular care patients, which was expected because of the different protocol (eight weeks distraction instead of six weeks) used in the OPS. The fact that the distraction duration in regular care is longer than in clinical trials when excluding the OPS is probably a result of the dependence on OR planning in regular care and the difference, being on average 2.7 days on 6-week protocol, was limited.

With an average intervention time of 53 minutes placing and 16 minutes removing the frame, the operative time is comparable to HTO and about half of the average time reported in literature for a TKA.[22–25] Complications were also described as similar to HTO and TKA [13–15], with pin tract infections, a common complication of external fixation in general [26], being the most prevalent complication in KJD. Complications of treatment were comparable between KJD patients treated in regular care and those treated in trials. With 70% of patients experiencing pin tract infections based on oral antibiotic use, they occurred more often than was previously seen in the RCTs, where around half of patients experienced infections.[13–15] This could be because in regular practice patients receive a standard antibiotics prescription and do not have to visit the hospital before starting their course, which makes it likely that

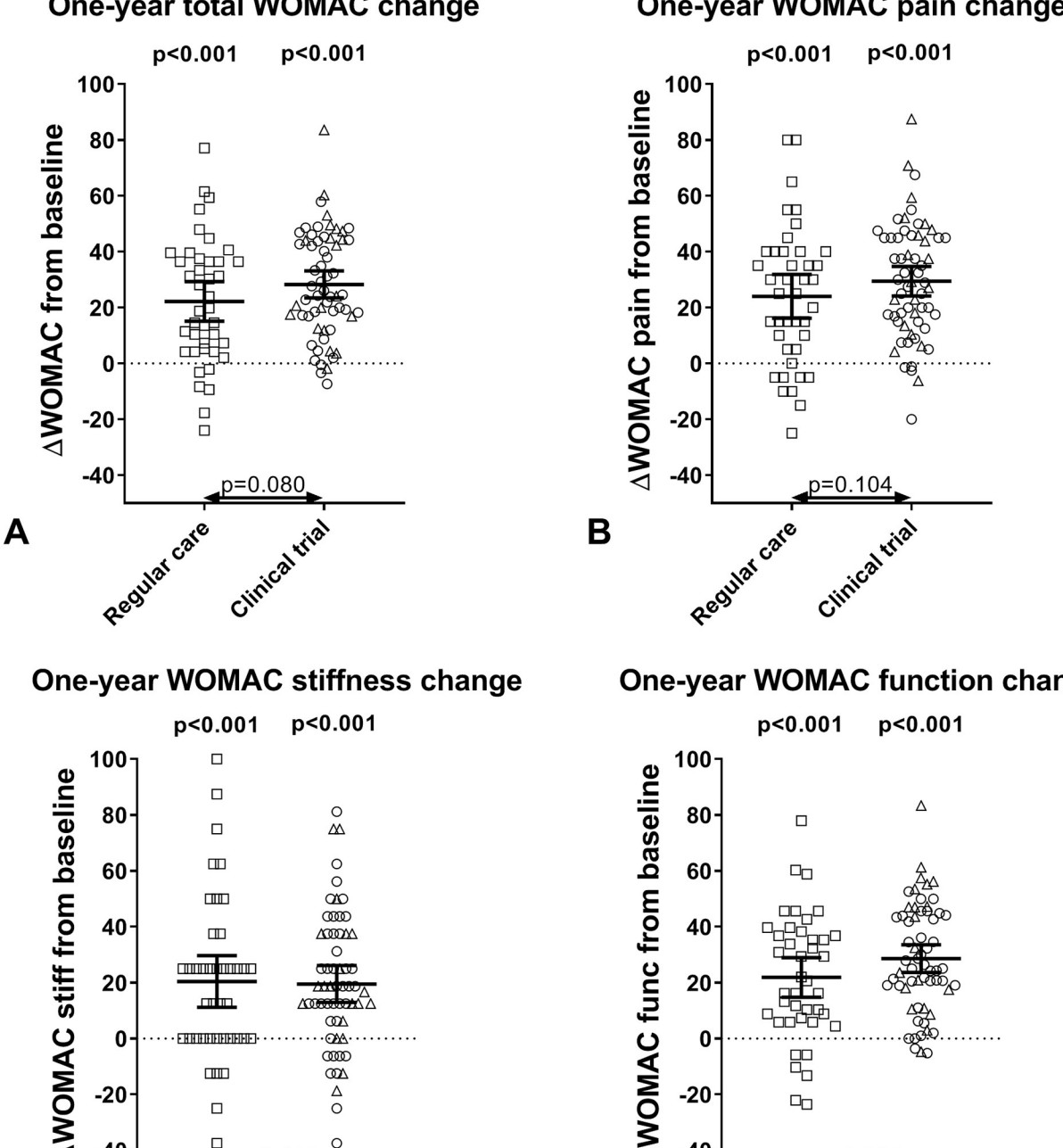

**Fig 4. One-year WOMAC change for patients treated with knee joint distraction.** One-year change in Western Ontario and McMaster Universities Osteoarthritis Index (WOMAC) total score (A) and the pain (B), stiffness (C) and function (D) subscales for patients treated with knee joint distraction in regular care and in OPS/RCT clinical trials (OPS = open prospective study; RCT = randomized controlled trial). P-values above groups indicate significant changes at one year compared to baseline while p-values between groups indicate the significance of differences between groups, corrected for baseline values and distraction duration. Each dot represents a patient (for trial patients: triangles represent OPS patients and circles RCT patients); bars represent mean and 95% confidence interval.

antibiotics are also used in case of doubtful infection. Pin tract infections had no significant influence on the clinical outcome at one year follow-up. Furthermore, despite the high occurrence of pin tract infections, patients undergoing TKA surgery several years after KJD have not experienced additional complications or diminished clinical efficacy.[17] Nevertheless, it is a major burden and effort should be made to reduce pin tract infections further. A new joint distraction device (KneeReviver) has been developed, which makes pin care easier. A clinical trial to evaluate this new device is currently ongoing. Additionally, new care protocols are encouraging, appearing to decrease the number of pin tracts significantly.

Not only pin tract infections, but complications in general did not significantly influence the clinical response. Complications other than pin tract infections did not occur with a frequency allowing statistical evaluation. However, the seventeen patients who received full KJD treatment in regular care and experienced other complications than pin tract infections all returned to the outpatient clinic after treatment and fourteen of them (82%) were satisfied with their KJD treatment and indicated that they had less OA complaints than before treatment. Only the other three patients (one who experienced pneumonia and flexion limitation, one a corpus liberum and one a broken bone pin) did not report success of the treatment. Clearly, there is room for improvement to decrease complications of the treatment to further improve the balance of benefit over burden.

A decrease in range of motion was seen as adverse effect previously in the clinical trials. In both regular care and clinical trials, the decrease that was seen shortly after KJD, recovered within months and normalized after a year, with the observed changes being minimal and less than the minimally detectable difference reported in literature.[27] As such, the differences are considered not to be clinically relevant and within variation of measurement.

The clinical benefit that was demonstrated previously in all clinical trials was also observed in regular care. In the clinical trials, the clinical benefit seemed slightly better, which was partly due to slightly better effects of the OPS treated patients. Although all not statistically significant, this may be the benefit of subtle differences in patient selection as well as the small difference in distraction duration (in favor of the OPS patients), as has been discussed before.[12] Moreover, no difference in the percentage of responders according to OMERACT-OARSI criteria at one year was observed either.

Neither being a patient from a clinical trial or regular care, nor any of the other baseline data predicted clinical outcome.

Unfortunately, while radiographs were performed in regular care to judge OA severity pre-treatment, a KLG of 2 or higher being a treatment prerequisite according to local guideline, these radiographs were not performed in standardized way, and neither were follow-up radiographs (amongst other including an aluminum step wedge for quantification of density and distances). Therefore JSW widening could not be quantified. In the three clinical trials, it has previously been shown that KJD causes a significant increase in radiographic JSW during the years after treatment, which has been related to cartilaginous tissue repair based on additional MRI evaluation and biochemical marker analyses.[8,9,13–15] Since no significant differences in patient characteristics and clinical benefit were found between regular care and trial patients, KJD in regular care may be expected to cause a similar structural response as supported by the representative pre- and post-treatment images shown.

This study had a number of limitations. First, around half of patients treated in regular care could not be used in the evaluation of clinical efficacy, as they did not fill out the questionnaires before and one year after treatment. As the regular care patients in this study were evaluated retrospectively, this could unfortunately not be solved. This might have caused a bias or misrepresentation of clinical results, although it was shown that the regular care patients who filled out the questionnaires did not differ in patient and treatment characteristics from those

who did not. Furthermore, for 93% of all regular care patients it is known they did not receive a TKA within a year, as they did attend the one-year outpatient clinic visit and/or filled out electronic questionnaires more than one year after treatment.

The second limitation of this study was that all regular care patients were treated in the same hospital. While other hospitals provide KJD treatment as well, they only started recently and clinical data was available only from our hospital. The patients from the clinical trials were treated in three different hospitals, however, and there were no statistically significant differences in patients' clinical benefit between these hospitals. This would therefore not be expected in regular care either.

This study did not include a control group of non-surgically treated patients. However, in the stage patients are considered for KJD they should be considered for TKA, but aged below 65 with persistent pain, a KLG of 2 or higher, and sufficient history of conservative treatment without sufficient success. As such, any good control group receiving no treatment would not be ethically sound for this population.

Despite the absence of statistically significant differences between patients treated in regular care and in clinical trials, patient selection and treatment conditions in regular care remain crucial for this novel joint saving treatment. The maximal effect regarding clinical benefit and structural repair has in all trials been obtained around one-year follow-up, sustaining for many years thereafter.[10,11] Therefore the one-year follow-up comparison with regular care outcome is considered predictive of the long-term outcome in regular care. Nevertheless, longer follow-up in regular care with larger number of patients is still warranted to proof this assumption. Moreover, such studies may benefit from standardized radiographs or MRI evaluation to evaluate joint tissue repair as well. Follow-up of more patients in regular care with proper data management may potentially provide treatment efficacy predictors, refining patient selection. Regardless, KJD as a regular care treatment results in significant clinical benefit one year post-treatment similar to that demonstrated in the clinical trials that have demonstrated sustainability of this initial effect. As such KJD, can be a joint-preserving of choice in relatively young patients with end stage knee OA.

## Author Contributions

**Conceptualization:** Simon C. Mastbergen, Ronald J. van Heerwaarden, Sander Spruijt, Floris P. J. G. Lafeber, Roel J. H. Custers.

**Data curation:** Mylène P. Jansen, Michelle D. van Empelen, Esmee C. Kester.

**Formal analysis:** Mylène P. Jansen, Michelle D. van Empelen.

**Funding acquisition:** Floris P. J. G. Lafeber.

**Investigation:** Ronald J. van Heerwaarden, Sander Spruijt, Roel J. H. Custers.

**Methodology:** Mylène P. Jansen, Simon C. Mastbergen, Floris P. J. G. Lafeber.

**Project administration:** Mylène P. Jansen, Michelle D. van Empelen.

**Resources:** Ronald J. van Heerwaarden, Sander Spruijt, Esmee C. Kester, Roel J. H. Custers.

**Supervision:** Simon C. Mastbergen, Floris P. J. G. Lafeber, Roel J. H. Custers.

**Writing – original draft:** Mylène P. Jansen.

**Writing – review & editing:** Simon C. Mastbergen, Ronald J. van Heerwaarden, Sander Spruijt, Michelle D. van Empelen, Esmee C. Kester, Floris P. J. G. Lafeber, Roel J. H. Custers.

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
