## [Decision Letter · Decision Letter 0]

14 Oct 2019

PONE-D-19-20825

Knee Joint Distraction in Regular Care for Treatment of Knee Osteoarthritis: A Comparison with Clinical Trial Data

PLOS ONE

Dear Ms Jansen,

Thank you for submitting your manuscript to PLOS ONE. After careful consideration, we feel that it has merit but does not fully meet PLOS ONE’s publication criteria as it currently stands. Therefore, we invite you to submit a revised version of the manuscript that addresses the points raised during the review process.

The authors are required to respond to the reviewers comments especially: 

The possible bias in the results of the study and how the authors tried to minimize this bias. The authors are required to discuss the complications with some more details about their management and how these complications had influenced the outcome. Adding more figures are required to show the effect of distraction. Also answering the question about the use of other radiographic tools, MRI, CT, etc, was there any place for such modalities in diagnosis or follow up of your cases?

We would appreciate receiving your revised manuscript by Nov 28 2019 11:59PM. To enhance the reproducibility of your results, we recommend that if applicable you deposit your laboratory protocols in protocols.io, where a protocol can be assigned its own identifier (DOI) such that it can be cited independently in the future. For instructions see: http://journals.plos.org/plosone/s/submission-guidelines#loc-laboratory-protocols

We look forward to receiving your revised manuscript.

Kind regards,

Osama Farouk

Academic Editor

PLOS ONE

Journal Requirements:

I have read the journal's policy and the authors of this manuscript have the following competings interests: FPJG Lafeber is co-founder, co-director, and shareholder of ArthroSave BV, a medical device company involved in marketing

a user-friendly knee joint distraction device.

Additional Editor Comments:

The authors are required to respond to the reviewers comments especially the possible bias in the results of the study and how the authors tried to minimize this bias. The authors are required to discuss the complications with some more details about their management and how these complications had influenced the outcome. Adding more figures are required to show the effect of distraction. Also answering the question about the use of other radiographic tools, MRI, CT, etc, was there any place for such modalities in diagnosis or follow up of your cases?

Reviewers' comments:

Reviewer's Responses to Questions

**Comments to the Author**

1. Is the manuscript technically sound, and do the data support the conclusions?

Reviewer #1: Yes

Reviewer #2: No

2. Has the statistical analysis been performed appropriately and rigorously? 

Reviewer #1: Yes

Reviewer #2: Yes

3. Have the authors made all data underlying the findings in their manuscript fully available?

Reviewer #1: No

Reviewer #2: No

4. Is the manuscript presented in an intelligible fashion and written in standard English?

Reviewer #1: Yes

Reviewer #2: Yes

5. Review Comments to the Author

Reviewer #1: Despite RCTs seem to be more scientifically sound than OPS, this article provides reader with additional scientific knowledge about this relatively new technique in treatment of Knee OA.

My comment is to do more thorough discussion about complications, not only infections, how did you treat, how these complications had influenced the outcome, etc.

Another comment is to add more figures to the paper to show the effect of distraction.

Third comment is about other radiographic tools, MRI, CT, etc, was there any place for such modalities in diagnosis or follow up of your cases?

Reviewer #2: the paper: Knee Joint Distraction in Regular Care for Treatment of Knee Osteoarthritis: A

Comparison with Clinical Trial Data presents no control group to objectively assess the outcome

There is risk of bias from the conflict of interest , the authors used a device they gain financial benefit from ( ArthroSave BV,)

the follow-up is very short to get to a conclusion that this device or this technique helps to delay TKR in OA knee

6. PLOS authors have the option to publish the peer review history of their article (what does this mean?). If published, this will include your full peer review and any attached files.

Reviewer #1: Yes: Ahmed H. K. Abdelaal

Reviewer #2: Yes: Prof. Dr Khaled M. Emara

---

## [Author Response · Author response to Decision Letter 0]

14 Nov 2019

We thank the reviewers for the helpful comments by which we consider the manuscript improved. Hopefully we answered all questions and addressed all comments to the intention and satisfaction of the reviewers.

Reviewer 1

- Review comment: 

My comment is to do more thorough discussion about complications, not only infections, how did you treat, how these complications had influenced the outcome, etc.

- Author response:

We agree that it is relevant to know how all complications were treated in regular care, not just focusing on the pin tract infections being the main complication. Therefore, in the revised Results section [line 220-235], we now mention for all types of complications how they were treated. While in the Results and the Discussion section we already included that experiencing complications did not have a significant influence on clinical benefit, we made this more explicit with the addition of the relevant p-value in the Results section [line 279-280]. Furthermore, we included a paragraph in the revised Discussion section [line 340-348] about the influence of complications other than pin tract infections on the clinical response. Complications other than pin tract infections did not occur in sufficient numbers to statistically test the influence of each complication on the clinical response, but we used the patients’ clinical experience as mentioned during their outpatient clinic visits after treatment to be able to elaborate on the potential effect of the complications on the clinical benefit. 

- Review comment: 

Another comment is to add more figures to the paper to show the effect of distraction.

- Author response:

We appreciate the comment, however, standardized radiographs or MRIs are not part of the protocol in in regular care, and as such are not available. This means we cannot objectively measure the structural effect of knee joint distraction in patients treated in regular care. In the three clinical trials we have previously demonstrated significant effect such as an increase in radiographic joint space width or improved cartilage thickness on MRI. We agree that the (structural) effect of distraction is interesting, and therefore added two examples (representatives of clear radiographic jointspace widening) of a trial and regular care patient [figure 2].

However, since data on structural repair are not available for regular care patients in quantitative measures and this outcome has already been published for trial patients, we feel that a focus on patient characteristics, complications and clinical effect should be the key message of the present manuscript. We referred to the added radiographs and included this point of discussion in the revised Discussion section of the manuscript [line 364-373]. 

- Reviewer comment: 

Third comment is about other radiographic tools, MRI, CT, etc, was there any place for such modalities in diagnosis or follow up of your cases?

- Author response:

Other radiographic tools, especially MRI, would indeed provide useful information in these patients. Unfortunately, no quantitative radiographic tools other than a non-standardized radiograph at inclusion demonstrating a Kellgren-Lawrence grade of at at least 2 (standard inclusion characteristic in the Dutch guidelines for knee joint replacement indication) and non-standardized radiographs post-treatement were used, because of the cost and the absence to use them for indication according to guidelines. Specifically for indication in future studies such imaging tools may be of help in regulare care to include the optimal population for this treatment, which is why we mention this shortly in the Discussion [line 402-403]. 

Reviewer 2:

- Reviewer comment: 

The article presents no control group to objectively assess the outcome.

- Author response:

The goal of this research was to compare patients treated in regular care to those treated in clinical trials, the latter being the control group for the regular care group. We agree that inclusion of patients that do not receive any treatment would be on interest. However, the population considered in this study comprise patients that present themselves in the hospital with a condition of end-stage knee OA needing surgical treatment, with in regular care indication total knee prosthesis but below the age of 65 years. As such any good control group receiving no or only conservative treatment, which all patients have had according to Dutch guidelines before they can be considered for arthroplasty, would not be ethical for this population. In the clinical trials, other control groups have previously been used: participants from the osteoarthritis initiative matched on patient characteristics with the KJD patients (Van der Woude et al, Cartilage 2017), and in the two randomized controlled trials patients treated with total knee arthroplasty (Van der Woude et al, Bone Joint J 2017; Jansen et al, Cartilage 2019) or high tibial osteotomy (Van der Woude et al, Knee 2017; Jansen et al, Cartilage 2019). Since such a comparion is not the key message of the present manuscript and these comparisons have been published already, we did not include them in this present manuscript. This choice has been added in the discussion section of the revised manuscript [line 290-394]. 

- Reviewer comment:

There is risk of bias from the conflict of interest, the authors used a device they gain financial benefit from (ArthroSave BV) .

- Author response:

This study is an investigator-initiated study and no industrial funding is involved, nor involvement in any orther way. Moreover, all patients in the trials and in regular care were treated with a proof of concept device and not with ArthroSave’s KneeReviver. ArthroSave’s KneeReviver was introduced less than a year ago, so there is still no sufficient follow-up. To avoid confusion, we removed reference to the company name (ArthroSave BV) from the revised Discussion section when referring to the new dedicated user-friendly frame provided by ArthroSave with limited occurrence of pin tract infections [line 337].

A risk of bias that we did recognize is the relatively high percentage of regular care patients that did not fill out both baseline and one-year follow-up questionnaires, as we acknowledged in the Discussion section mentioning how we tried to minimize this bias [line 375-383]. 

- Reviewer comment: 

The follow-up is very short to get to a conclusion that this device or this technique helps to delay TKR in OA knee

- Author response: 

We agree that we cannot yet draw a conclusion about whether or not KJD treatment can postpone TKR in regular care after this one year of follow-up. However, all trials have demonstrated the long-term clinical and structural benefit obtained (e.g. Jansen et al OAC 2018 Dec;26(12):1604-1608). The maximum (optimum) effects in all these studies was around 1 year, sustaining for years thereafter. As such we feel that this one-year follow-up comparison is indicative for long-term follow-up as well. However as this needs confirmation in regular care we rephrased the final sentence of the revised manuscript [line 405-409].

---

## [Decision Letter · Decision Letter 1]

13 Dec 2019

PONE-D-19-20825R1

Knee Joint Distraction in Regular Care for Treatment of Knee Osteoarthritis: A Comparison with Clinical Trial Data

PLOS ONE

Dear Ms Jansen,

Thank you for submitting your manuscript to PLOS ONE. After careful consideration, we feel that it has merit but does not fully meet PLOS ONE’s publication criteria as it currently stands. Therefore, we invite you to submit a revised version of the manuscript that addresses the points raised during the review process.

We would appreciate receiving your revised manuscript by Jan 27 2020 11:59PM. To enhance the reproducibility of your results, we recommend that if applicable you deposit your laboratory protocols in protocols.io, where a protocol can be assigned its own identifier (DOI) such that it can be cited independently in the future. For instructions see: http://journals.plos.org/plosone/s/submission-guidelines#loc-laboratory-protocols

We look forward to receiving your revised manuscript.

Kind regards,

Osama Farouk

Academic Editor

PLOS ONE

Reviewers' comments:

Reviewer's Responses to Questions

**Comments to the Author**

1. If the authors have adequately addressed your comments raised in a previous round of review and you feel that this manuscript is now acceptable for publication, you may indicate that here to bypass the “Comments to the Author” section, enter your conflict of interest statement in the “Confidential to Editor” section, and submit your "Accept" recommendation.

Reviewer #1: All comments have been addressed

Reviewer #3: (No Response)

2. Is the manuscript technically sound, and do the data support the conclusions?

Reviewer #1: Yes

Reviewer #3: Yes

3. Has the statistical analysis been performed appropriately and rigorously? 

Reviewer #1: Yes

Reviewer #3: Yes

4. Have the authors made all data underlying the findings in their manuscript fully available?

Reviewer #1: Yes

Reviewer #3: Yes

5. Is the manuscript presented in an intelligible fashion and written in standard English?

Reviewer #1: Yes

Reviewer #3: Yes

6. Review Comments to the Author

Reviewer #1: (No Response)

Reviewer #3: Dear author: thank you much for sending this important topic article as I personally believe that KJD will gain more popularity in the future.

However, you need to follow the following points before possible accepting your article for publication:

1- Remove the name of your hospitak UMC from all pages of the article. You can write: our hospital Instead.

2- in line 97, remove the country name Netherland and hospital name....remove all the Line as no need.

3- in table 2, what do you mean by Corpus liberum ? and why breaking of bone pin happen ?

4- in discussion: you referred the high pin tract infection in the trial group to not taking antibiotic regularly as happened in regular practice group !

But actually, there are many factors influence this! The rate of pin tract infection in publications range from 0-100% ! This means that it is 0% in hands of some one who follow some tricks while it is 100% in some other hands ! To improve that, a proper new drill bit with the right diameter should be chosen and proper drill speed used and a soothing saline should be used not a dry drilling technique! A proper 6 mm. Diameter pins should be used and there should not be any stress to any pin in the Ex Fix. As any stress will be reflected to the bone and leads to loosening abs infection.

5- in line 308, you pointed to a new device used in the study: arthrosave.

Did you use in all patients ? Please add a figure for a photo of the new device to the study.

6- please add a brief surgical technique to the study as it is valuable for the reader to know how to do KJD in the future.

7- The X Ray in figure 1, does not show clearly the planes of pins ! Are they pure medial and lateral ? Kindly describe.

8- I can see 2 arthrosave devices were applied in the sane fig. Please explain why 2 devices and not one is enough ?

9- in the same fig. I can see proximal clamp on the medial side showing a mechanical problem of the pins as pins are divergent. The distance between the 2 pins are closer at the clamp level while it is longer at the bone level. It is clear that the distal pin is blended. Actually this leads to a big stress to the bone cortex and can leads to many complications as pin tract infection, loosening, pin breakage or bone breaking as you mentioned 2 cases with this problem among the trial group which is about 5% ! This is really much and if the pins were inserted properly without any stress then the complication rate will be much less.

I would like if you can choose another X Ray showing the device with distraction with a proper technique as many young specialists colleagues will see your fig. In the future and try to copy so we must provide them with the right technique. I am sure that you can choose better X Ray to show.

10- what are the pin diameter you used ? Please write this in the study as it is important to know. Also write it for the regular care group.

11- I can see a long tapered tip of the pins that are not introduced well to pass the far cortex of the bone. This could be a direct cause of instability and loosening and infection .

Please explain that or remove this fig and choose a better one with the proper technique to give a good example to the reader .

7. PLOS authors have the option to publish the peer review history of their article (what does this mean?). If published, this will include your full peer review and any attached files.

Reviewer #1: Yes: Ahmed H.K. Abdelaal

Reviewer #3: Yes: Yasser Elbatrawy, Professor of Orthopedic surgery, Al-Azhar university, Cairo, Egypt.

---

## [Author Response · Author response to Decision Letter 1]

16 Dec 2019

We thank the reviewer for the helpful comments by which we consider the manuscript improved. Hopefully we answered all questions and addressed all comments to the intention and satisfaction of the reviewer.

- Review comment: 

Remove the name of your hospital UMC from all pages of the article. You can write: our hospital instead.

- Author response:

As suggested, we removed the hospital name ‘UMC Utrecht’. We either replaced it with ‘our hospital’ [lines 28, 74, 164, 197, 399] or completely removed it [line 398] when appropriate. However, we did not remove ‘University Medical Center Utrecht’ when referring to the ethical review committee in the methods section [line 98/99] because this refers not to the hospital but to the medical ethical review committee that approved the studies (as required to mention according to the PLOS ONE submission guidelines).

- Reviewer comment:

In line 97, remove the country name Netherland and hospital name....remove all the Line as no need. 

- Author response:

As suggested, we removed any reference to The Netherlands (‘Dutch’) in [line 24, 66, 398] and replaced ‘Dutch’ with ‘local’ in [line 75, 377]. However, we did not remove ‘Netherlands Trial Register’ in the methods section [line 99/100] because this is the official name of where the trials were registered and can be found online (as required to mention according to the PLOS ONE submission guidelines).

- Reviewer comment:

In table 2, what do you mean by Corpus liberum? and why breaking of bone pin happen?

- Author response: 

The corpus liberum was a loose piece of cartilage/bone present in the knee. This short explanation has been added in [line 246]. It is unfortunately not known why the bone pin broke, and as we do not want to speculate on this without being sure, we added ‘reason unknown’ in the complications section of the revised manuscript [line 233].

- Reviewer comment: 

In discussion: you referred the high pin tract infection in the trial group to not taking antibiotic regularly as happened in regular practice group! But actually, there are many factors influence this! The rate of pin tract infection in publications range from 0-100%! This means that it is 0% in hands of someone who follow some tricks while it is 100% in some other hands! To improve that, a proper new drill bit with the right diameter should be chosen and proper drill speed used and a soothing saline should be used not a dry drilling technique! A proper 6 mm. Diameter pins should be used and there should not be any stress to any pin in the Ex Fix. As any stress will be reflected to the bone and leads to loosening abs infection.

- Author response: 

We agree that there are many factors that affect the amount of pin tract infections. As mentioned in the results section [line 228], a new wound care protocol already managed to decrease the amount of pin tract infections from 85% in the first clinical trial (the open prospective study) to 57% in the following clinical trials (the randomized controlled trials). In the regular care group the percentage of antibiotic use was with 70% a bit higher than in the RCTs and we hypothesize that this increased percentage may be because the infections are not diagnosed in person by a physician, but patients always receive a standard prescription of oral antibiotics to take in case of suspected pin tract infections. As such, the amount of patients taking antibiotics is not necessarily the same as the amount of patients experiencing pin tract infections. Nevertheless, we should aim to decrease the amount of pin tract infections and antibiotics courses as much as possible by indeed making appropriate changes to the regular care protocol, which we are doing by using the newly developed distraction device and evaluating new regular care protocols, as mentioned in the discussion [line 348-351]. We will also take your suggestions into account in future adjustments to hopefully decrease the amount of pin tract infections.

- Reviewer comment: 

In line 308, you pointed to a new device used in the study: ArthroSave. Did you use in all patients? Please add a figure for a photo of the new device to the study.

- Author response:

The mention of the company name ‘ArthroSave’ was removed in the previous revised version of the manuscript on request of the other reviewers, in order to remove any potential conflict of interest regarding the new device. This newer device (the KneeReviver) was not used on any of the patients used in this manuscript and is currently being analyzed in an ongoing clinical study. We mention it in the discussion only because this new device is specifically designed to decrease pin tract infections, as these infections increase the patient’s treatment burden. However, as none of the patients in this study were treated with this device, we did not add a photo of this new device to avoid confusion. 

- Reviewer comment:

Please add a brief surgical technique to the study as it is valuable for the reader to know how to do KJD in the future.

- Author response:

We added some extra information regarding the surgical technique in the revised version of the manuscript [line 105-116].

- Reviewer comment: 

The X Ray in figure 1, does not show clearly the planes of pins! Are they pure medial and lateral? Kindly describe.

- Author response: 

The planes of the pins can indeed not be accurately visualized in the radiograph in figure 1. The pins are not positioned purely medially and laterally and an explanation of how the pins should be positioned was included in a brief description of the surgical technique. We added some extra information regarding the surgical technique in the revised version of the manuscript [line 105-116].

- Reviewer comment: 

I can see 2 arthrosave devices were applied in the sane fig. Please explain why 2 devices and not one is enough?

- Author response:

One distraction device consists of two separate tubes that are both fixed to the femur and tibia with bone pins, one laterally and one medially, as highlighted in the methods sections [line 105-106]. The distraction treatment is always performed on both sides of the knee, and as such with two tubes, so that both the medial and lateral side are distracted properly. A unilateral frame with one tube provides insufficient distraction to the contralateral compartment because of a certain degree of flexibility in the distraction frame and pins.

- Reviewer comment:

In the same fig. I can see proximal clamp on the medial side showing a mechanical problem of the pins as pins are divergent. The distance between the 2 pins are closer at the clamp level while it is longer at the bone level. It is clear that the distal pin is blended. Actually this leads to a big stress to the bone cortex and can leads to many complications as pin tract infection, loosening, pin breakage or bone breaking as you mentioned 2 cases with this problem among the trial group which is about 5%! This is really much and if the pins were inserted properly without any stress then the complication rate will be much less. I would like if you can choose another X Ray showing the device with distraction with a proper technique as many young specialists colleagues will see your fig. In the future and try to copy so we must provide them with the right technique. I am sure that you can choose better X Ray to show.

- Author response:

We agree that pin positioning is very important and the goal is to position the pins perpendicularly instead of divergent. It is therefore indeed better to choose a radiograph with perpendicular and parallel positioning and replaced figure 1 with an image where the pins are positioned fully correct. 

- Reviewer comment:

What are the pin diameter you used? Please write this in the study as it is important to know. Also write it for the regular care group.

- Author response:

We agree that this is a detail that should be included. As mentioned we extended the brief explanation on the used surgical technique [line 105-116] and in this part we also included the pin diameter (5mm). As mentioned in the methods section of the manuscript [line 142] the treatment procedure was identical for the trial group and regular care group. 

- Reviewer comment:

I can see a long tapered tip of the pins that are not introduced well to pass the far cortex of the bone. This could be a direct cause of instability and loosening and infection. Please explain that or remove this fig and choose a better one with the proper technique to give a good example to the reader.

- Author response: 

It is indeed important that the pins properly pass the far cortex of the bone, as described in the extended surgical technique description in the revised methods section [line 105-117]. However, the radiograph is performed in AP direction and the pins of especially the tibia are positioned under an angle and not straightly in medial-lateral direction, which means the pins protrude to the second cortex of the bone slightly on the posterior side. Because of this, the AP radiograph do not show the pins exit the bone, even though they do.

---

## [Decision Letter · Decision Letter 2]

6 Jan 2020

Knee Joint Distraction in Regular Care for Treatment of Knee Osteoarthritis: A Comparison with Clinical Trial Data

PONE-D-19-20825R2

Dear Dr. Jansen,

We are pleased to inform you that your manuscript has been judged scientifically suitable for publication and will be formally accepted for publication once it complies with all outstanding technical requirements.

With kind regards,

Osama Farouk

Academic Editor

PLOS ONE

Additional Editor Comments (optional):

Reviewers' comments:

Reviewer's Responses to Questions

**Comments to the Author**

1. If the authors have adequately addressed your comments raised in a previous round of review and you feel that this manuscript is now acceptable for publication, you may indicate that here to bypass the “Comments to the Author” section, enter your conflict of interest statement in the “Confidential to Editor” section, and submit your "Accept" recommendation.

Reviewer #1: All comments have been addressed

Reviewer #3: All comments have been addressed

2. Is the manuscript technically sound, and do the data support the conclusions?

Reviewer #1: Yes

Reviewer #3: Yes

3. Has the statistical analysis been performed appropriately and rigorously? 

Reviewer #1: Yes

Reviewer #3: Yes

4. Have the authors made all data underlying the findings in their manuscript fully available?

Reviewer #1: Yes

Reviewer #3: Yes

5. Is the manuscript presented in an intelligible fashion and written in standard English?

Reviewer #1: Yes

Reviewer #3: Yes

6. Review Comments to the Author

Reviewer #1: (No Response)

Reviewer #3: Dear author: thank you for your revision.

Congratulation.

From an expert point of view: I like to advise only 2 things:

1- Always use of 6 mm. Pins.

2- Never use self tapping self drilling pins as they have always short longevity in bone and tend to get loosening and pin tract infection earlier than when using drilling and classic HA coated pins ( Hydroxyl Apatite ).

7. PLOS authors have the option to publish the peer review history of their article (what does this mean?). If published, this will include your full peer review and any attached files.

Reviewer #1: Yes: Ahmed H. K. Abdelaal

Reviewer #3: Yes: Professor dr. Yasser Elbatrawy

---

## [Editor Report · Acceptance letter]

9 Jan 2020

PONE-D-19-20825R2 

Knee Joint Distraction in Regular Care for Treatment of Knee Osteoarthritis: A Comparison with Clinical Trial Data 

Dear Dr. Jansen:

I am pleased to inform you that your manuscript has been deemed suitable for publication in PLOS ONE. Congratulations! Your manuscript is now with our production department. 

With kind regards,

on behalf of

Dr. Osama Farouk 

Academic Editor

PLOS ONE